# Accelerating Deep Learning using Ivy

**Guillermo Sanchez-Brizuela**
Unify.ai
guillermo@unify.ai

**Ved Patwardhan**
Unify.ai
ved@unify.ai

**Matthew Barrett**
Unify.ai
matthew@unify.ai

**Paul Anderson**
Unify.ai
paul@unify.ai

**Mustafa Hani**
Unify.ai
mustafa@unify.ai

**Daniel Lenton**
Unify.ai
dan@unify.ai

## Abstract

Today's machine learning (ML) ecosystem suffers from deep fragmentation due to the proliferation of numerous incompatible frameworks, compiler infrastructure and hardware. Each unique tool within this fragmented stack has its own set of benefits and drawbacks, making it better suited for certain use-cases. As a result, different areas of industry and academia use different tools for different use cases, which hinders collaboration and democratization, ultimately resulting in costly re-implementations and sub-optimal runtime efficiency when deploying, due to sparse and partial connections to the rest of the stack. In this paper, we present Ivy, a complementary, multi-backend ML framework, and its transpiler, which aims to bridge this gap and solve the fragmentation problem by enabling the integration of code from one framework into another to speed up research, development, and model inference.

## 1 Introduction

Machine learning (ML) has advanced significantly over the past decade. This has inspired the developer community to build many open-source tools that accelerate the research, development and deployment of ML applications. TensorFlow [1], PyTorch [2] and JAX [3] have emerged as front-runners for ML, while NumPy [4] is considered the standard framework for numerical computation.

Different users have diverging preferences based on their use-case. For example, PyTorch is preferred by many researchers for its ease of use, Pythonic design and dynamic graphs, others prefer TensorFlow due to its efficiently compiled graphs, superior deployment features, and bindings for edge and mobile devices, while some prefer JAX due to its fully functional form, superior runtime efficiency on TPUs, and unmatched flexibility for gradient computation.

Ivy aims to solve this problem by enabling interoperability between these frameworks, making code portable and reusable. By doing so, Ivy not only democratizes ML, as every tool and model can now be used by developers using other frameworks, but it also unlocks framework-specific infrastructure and support for hardware that wasn't available before, accelerating development, training and inference of ML models.

Exchange formats [5, 6, 7] have eased fragmentation at the lower levels. For example, ONNX was established to facilitate cross-framework compatibility by defining a standard set of operators that can be used across different hardware targets. However, it mainly focuses on core neural network operations and has limited support for other general array processing functions. Some frameworks have added the ability to import or export ONNX models, allowing for some level of

Workshop on Advancing Neural Network Training at 37th Conference on Neural Information Processing Systems (WANT@NeurIPS 2023).

model conversion between frameworks, but it is primarily built for deployment rather than training. Additionally, framework maintainers must implement their own bindings to ONNX, which they don't always fully maintain. In contrast, Ivy proactively binds to all frameworks, without any need for framework creators to coalesce around our standard.

In this paper, we present the Ivy framework, a set of APIs that enable the development of framework-agnostic code and serves as an intermediate representation between frameworks, and the Ivy transpiler, a tool capable of converting code between frameworks.

## 2   Ivy

The Ivy project is composed of various components that collectively make interoperability of code between different ML frameworks possible.

### 2.1   The Ivy framework

The Ivy Framework attempts to provide an interface that enables unified behavior across the target backends. The framework is composed of three *unique* APIs plus the backend API, which replicates the Ivy API with framework-specific implementations (Figure 1).

**Functional API.** The framework revolves around a unified functional API that is the lowest level of abstraction provided. The functional API supports multiple backends, namely NumPy, JAX, TensorFlow, PyTorch and PaddlePaddle, each one with a backend-specific implementation of the Ivy functional API. This functional API implements all functions in the Array API standard [8] which contains the most commonly used array-processing functions, as well as many others not in the standard.

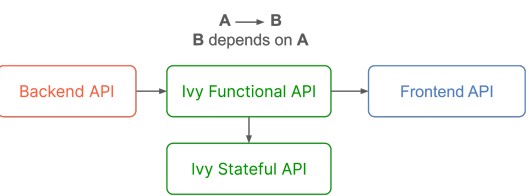

Figure 1: Dependency graph of the main building blocks and APIs in the Ivy framework.

**Stateful API.** Ivy also has a stateful API, which provides high-level modules and optimizers for rapid prototyping. The stateful API is built on top of the functional API and is therefore backend-agnostic as well.

**Frontend API.** Frontend APIs have been implemented for PyTorch, TensorFlow, NumPy, JAX and PaddlePaddle. The goal of a frontend API is to replicate the behaviour of each function in that framework's functional API with an implementation that only uses functions from Ivy's functional API. Since Ivy's functional API can be executed using any of the supported backends, we can therefore use these frontends to reproduce the original behaviour of a framework, but using any of the supported backends. This way, Ivy's functional API acts as an Intermediate Representation (IR) between frameworks.

### 2.2   The Ivy transpiler

Code transpilation refers to the process of converting code written in one framework to a different one or a different version of the source framework. Examples of this procedure would be porting PyTorch code to JAX for faster execution, converting JAX models to PyTorch in order to integrate them into existing pipelines or build on top of them, or seamlessly updating versions when breaking changes occur.

A fundamental part of the transpiler is the graph tracer, which extracts a computational graph of any array-computing-based function or trainable module. Internally, Ivy's tracer registers any functional-API level call that occurs during a given computing procedure alongside relevant information from that particular function call, mainly, unique identifiers of its inputs and outputs. Once this information has been recorded, the tracer reconstructs a directed acyclic graph (DAG) recursively, starting from the output of the top-level system and connecting the inputs and outputs of every function that contributes mathematically to the computation.

Leveraging the graph tracer and Ivy's frontends and backends APIs, Ivy enables code conversion in the transpiler following the process outlined in Figure 2:

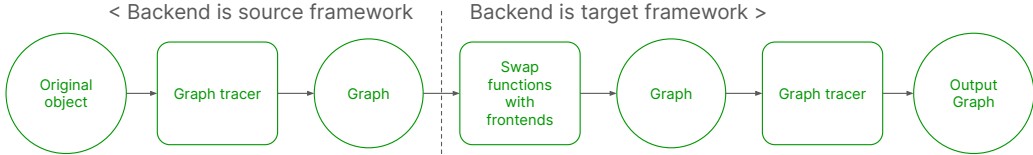

Figure 2: Overview of the transpilation process and its intermediate stages.

1. First tracing pass: We call the graph tracer a to generate a fully functional computational graph that captures the operations of the function/module. The main objective of this step is to decompose the original system, including its high-level classes, into fundamental building blocks from the functional API of the source framework.

2. Replacement of the functional-API level nodes and conversion of parameters: Once we have a fully functional graph of the system, we can leverage Ivy's frontend APIs and backend APIs to perform code conversion. As the frontends mirror the signatures of the frameworks' functional API, we can replace the original functions of the nodes in the captured graph with their Ivy frontend equivalent. As the frontends are implemented using Ivy's functional API, we effectively link the original code to the corresponding functions in any other framework available in the backends, since we can change the framework used to execute Ivy functions. During this replacement stage, we also convert cached and non-tracked objects from the source to the target framework, including constant arrays, native data types or device-classes.

3. Second tracing pass: Although at this stage the graph is fully composed of Ivy functions, and thus already fully interoperable with the target framework, there is some overhead due to the functions called in the frontends and the functional API of Ivy. To remove this overhead entirely and generate an efficient graph of pure operations in the target framework, we trace the system once again. By doing so, we remove the wrapping layers from Ivy's frontends and Ivy's functional API and extract the underlying functions from the target framework functional API.

Once the graph has been transpiled, it's fully composed of functions from the target framework, with all trainable parameters already converted to the corresponding native objects by Ivy. This ensures that all operations are differentiable and that native optimizations and tools are instantly available, regardless of the original framework. As a result, Ivy empowers the user to train and fine-tune the converted models using their preferred framework-specific training loops and infrastructure.

## 3    Use cases of the transpiler

As previously outlined, Ivy's transpiler can be used for different purposes. In this section, we will go over its main value propositions.

### 3.1    Libraries and tools interoperability

As a direct consequence of the coexistence of different frameworks, different ecosystems of libraries and tools have been developed around them. This means that developers are often limited to using the tools already available in their corresponding ecosystem. Ivy can help in two different ways here. The first one is by transpiling an entire array-based library to a different framework. This way, all functions contained within the library will be converted to the target framework, which can then be used as usual. Secondly, Ivy can also enable the user to leverage training tools, pipelines, and general infrastructure that has been designed for a specific framework models, simply by converting the model from the user framework to the one required by the library.

Table 1: Latency improvement of a collection of HF models after being transpiled from PyTorch to JAX when compared to the original model compiled using PyTorch Dynamo with the TorchInductor backend and different compilation modes.

| Model | Over Default | Over Reduce Overhead | Over Max Autotune | Over Best PyTorch Mode |
|---|---|---|---|---|
| BEiT [9] | 1.79 | 1.18 | 1.14 | 1.14 |
| BERT [10] | 3.88 | 1.18 | 1.17 | 1.17 |
| BiT [11] | 3.45 | 1.14 | 1.11 | 1.11 |
| CamemBERT [12] | 4.20 | 1.19 | 1.18 | 1.18 |
| ConvNeXt [13] | 3.04 | 1.18 | 1.08 | 1.08 |
| CvT [14] | 4.94 | 1.21 | 1.17 | 1.17 |
| DeiT [15] | 1.79 | 1.20 | 1.12 | 1.12 |
| DistilBERT [16] | 3.83 | 1.07 | 1.06 | 1.06 |
| MobileNet-v2 [17] | 9.70 | 1.89 | 1.86 | 1.86 |
| ResNet [18] | 4.03 | 1.37 | 1.43 | 1.37 |
| ViT [19] | 1.68 | 1.14 | 1.12 | 1.12 |
| MobileBERT [20] | 6.68 | 0.93 | 0.98 | 0.93 |
| RoBERTa [21] | 4.22 | 1.20 | 1.19 | 1.19 |
| Roformer [22] | 4.94 | 1.27 | 1.26 | 1.26 |
| Swin-v2 [23] | 5.91 | 3.42 | 3.88 | 3.42 |
| Whisper [24] | 1.60 | 1.08 | 1.01 | 1.01 |
| BlenderBot [25] | 3.08 | 0.79 | 0.79 | 0.79 |
| Wav2vec2 [26] | 2.20 | 1.20 | 1.14 | 1.14 |
| CLIP [27] | 3.67 | 0.99 | 0.98 | 0.98 |

A code snippet containing the usual workflow for library transpilation is available in Appendix 5.1, where we transpile the Kornia library from PyTorch to JAX.

## 3.2 Integration of trainable modules

Trainable modules, such as specific layers or models, are originally published for one particular framework. This results in costly reimplementations where details can go unnoticed, hampering the reproducibility of the results. In this case, Ivy can automatically convert these trainable modules to accelerate research and bring the latest models to every developer. As the outcome of the transpilation process in Ivy is composed of functions from the functional API of the target framework, this means that the output graph (in this case, contained a trainable module in the target framework) is fully differentiable and can be trained or fine-tunes as if it had been written in the target framework from scratch.

Appendix 5.2 includes an example where a Haiku module is transpiled to PyTorch and then used as the backbone for a classifier, showing how this allows the user to bring any module or section of it to their preferred framework and then build on top of it to obtain a trainable module that mimics the computational behavior of the original system.

## 3.3 Training and inference acceleration

The previous subsection showcases how a transpiled module from framework A to B can be integrated into another model in framework B to accelerate research and the development of new models. However, the transpilation of a trainable module by itself can also be beneficial in terms of computational resources. We can transpile a model to leverage optimizations that are only available in certain compilers or to enable the use of hardware which is not as well supported in the source framework. As an example of this, we include the results of a set of benchmarks performed on a selected set of models from the Hugging Face's Transformers library where we compare the original PyTorch model and the transpiled version in JAX.

In Table 1, we can see how even after the latest improvements in the PyTorch compiling tools, which accelerate PyTorch code significantly, the conversion of PyTorch models to JAX still bring a mean improvement in the execution speed of the model of $1.27\times$, mainly due to the different optimizations happening at the XLA level. For this experiment, we average the latency over 100 runs in a warmed-up NVIDIA V100 GPU. Table 2 in Appendix 5.3 includes the measurements used to calculate the increments.

# 4 Conclusions

In this paper we have presented Ivy, a multi-backend ML framework, and its transpiler. Through the transpiler, Ivy allows developers to integrate code from any framework into another, removing the friction associated with reimplementations and enables the use of any library, tool, or infrastructure that is better supported in a different frameworks, including specialized compilers and devices. Consequently, Ivy has the potential to accelerate any kind of ML-related research or development process, model training, or deployed system.

Moreover, we are currently working on integrating state of the art tools to perform model compression, quantization and tensorization natively as part of Ivy, as well as specialized low-level optimizations such as efficient, hardware-specific kernels. Through the combination of these framework-specific SOTA tools and the various compilation pipelines that transpilation unlocks, we foresee synergies that will further improve latency and throughput on all kinds of ML models.

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

# 5 Appendices

## 5.1 Library transpilation

The following code snippet includes an example on how to perform code transpilation using Ivy to convert the Kornia library, originally written in PyTorch, to JAX.

```python
import ivy
import cv2
import kornia
import tensorflow as tf

# transpile module lazily
kornia = ivy.transpile(kornia, to="tensorflow")

# load image into tensorflow
img = tf.constant(cv2.imread("image.png")) / 255
img = tf.expand_dims(tf.transpose(img, (2, 0, 1)), 0)

# custom composite
def dilate_edges(img):
    edges = kornia.filters.canny(img)[1]
    return kornia.morphology.dilation(edges, tf.ones(7, 7))

# dilate edges
new_img = dilate_edges(img) # slow, transpiling
new_img = dilate_edges(img) # fast, transpiled
```

## 5.2 Model transpilation

In a similar fashion, the following code showcases how to transpile a Haiku model to PyTorch.

```python
import ivy
import jax
import torch

# Get a pretrained haiku model
from deepmind_perceiver_io import key, perceiver_backbone

# Transpile it into a torch.nn.Module with the corresponding parameters
dummy_input = jax.random.uniform(key, shape=(1, 3, 224, 224))
params = perceiver_backbone.init(rng=key, images=dummy_input)
backbone = ivy.transpile(
    perceiver_backbone, to="torch", params_v=params, kwargs={"images": dummy_input}
)

# Build a classifier using the transpiled backbone
class PerceiverIOClassifier(torch.nn.Module):
    def __init__(self, num_classes=20):
        super(PerceiverIOClassifier, self).__init__()
        self.backbone = backbone
        self.max_pool = torch.nn.MaxPool2d((512, 1))
        self.flatten = torch.nn.Flatten()
        self.fc = torch.nn.Linear(1024, num_classes)

    def forward(self, x):
        x = self.backbone(images=x)
        x = self.flatten(self.max_pool(x))
        return self.fc(x)

# Initialize a trainable, customizable, torch.nn.Module
classifier = PerceiverIOClassifier()
ret = classifier(torch.rand((1, 3, 224, 224)))
```

## 5.3 Latency measurements

The following table presents the latency results from the experiment described in Section 3.3. As stated, these measurements have been used to calculate the increments shown in Table 1.

Table 2: Latency measurements when running a collection of HF models using the default compilation method in PyTorch Dynamo, PyTorch compilation using the Reduce Overhead and the Max Autotune options, and the model after being transpiled from PyTorch to JAX. Results are given in milliseconds.

| Model | Default Dynamo | Reduce Overhead | Max Autotune | JAX |
|---|---|---|---|---|
| BEiT [9] | $10.20 \pm 1.14$ | $6.76 \pm 0.28$ | $6.48 \pm 0.43$ | $5.70 \pm 1.44$ |
| BERT [10] | $8.70 \pm 0.73$ | $2.61 \pm 0.07$ | $2.58 \pm 0.11$ | $2.20 \pm 0.57$ |
| BiT [11] | $10.03 \pm 0.58$ | $3.32 \pm 0.24$ | $3.22 \pm 0.08$ | $2.91 \pm 0.84$ |
| CamemBERT [12] | $9.16 \pm 1.11$ | $2.60 \pm 0.06$ | $2.57 \pm 0.04$ | $2.17 \pm 0.54$ |
| ConvNeXt [13] | $8.00 \pm 0.59$ | $3.10 \pm 0.14$ | $2.85 \pm 0.08$ | $2.63 \pm 0.75$ |
| CvT [14] | $19.52 \pm 0.93$ | $4.78 \pm 0.46$ | $4.61 \pm 0.47$ | $3.95 \pm 0.69$ |
| DeiT [15] | $10.18 \pm 1.26$ | $6.82 \pm 0.43$ | $6.37 \pm 0.24$ | $5.68 \pm 1.42$ |
| DistilBERT [16] | $4.53 \pm 0.54$ | $1.27 \pm 0.05$ | $1.24 \pm 0.04$ | $1.18 \pm 0.28$ |
| MobileNet-v2 [17] | $9.61 \pm 0.97$ | $1.88 \pm 0.20$ | $1.83 \pm 0.08$ | $0.99 \pm 0.10$ |
| ResNet [18] | $9.74 \pm 0.79$ | $3.30 \pm 0.16$ | $3.44 \pm 0.85$ | $2.41 \pm 0.74$ |
| ViT [19] | $9.54 \pm 1.18$ | $6.51 \pm 0.12$ | $6.36 \pm 0.39$ | $5.68 \pm 1.49$ |
| MobileBERT [20] | $34.57 \pm 3.84$ | $4.84 \pm 0.11$ | $5.07 \pm 0.66$ | $5.17 \pm 0.68$ |
| RoBERTa [21] | $9.24 \pm 0.82$ | $2.63 \pm 0.08$ | $2.60 \pm 0.06$ | $2.18 \pm 0.55$ |
| Roformer [22] | $10.89 \pm 0.70$ | $2.80 \pm 0.34$ | $2.76 \pm 0.07$ | $2.20 \pm 0.73$ |
| Swin-v2 [23] | $22.00 \pm 1.05$ | $12.72 \pm 0.95$ | $14.45 \pm 1.24$ | $3.72 \pm 0.98$ |
| Whisper [24] | $10.95 \pm 1.05$ | $7.36 \pm 0.25$ | $6.89 \pm 0.13$ | $6.84 \pm 1.63$ |
| BlenderBot [25] | $21.47 \pm 0.72$ | $5.47 \pm 0.12$ | $5.48 \pm 0.18$ | $6.96 \pm 1.38$ |
| Wav2vec2 [26] | $1.16 \pm 1.12$ | $6.32 \pm 0.29$ | $6.01 \pm 0.03$ | $5.24 \pm 4.26$ |
| CLIP [27] | $20.70 \pm 1.67$ | $5.40 \pm 0.18$ | $5.35 \pm 0.92$ | $5.46 \pm 0.97$ |

