# OpenReview forum: "Accelerating Deep Learning using Ivy"
_NeurIPS.cc/2023/Workshop/WANT — WANT@NeurIPS 2023 Poster_

### Official Review · Reviewer_Yxqk · 2023-10-25
**Bridging the Fragmented DL Landscape**

**Confidence:** 5

**Review:**

This paper introduces Ivy, a multi-backend framework designed to address the fragmentation in the current machine-learning ecosystem caused by many incompatible tools and frameworks. It includes a transpiler integrating code from one framework into another, accelerating research, development, and model inference. The primary goal of Ivy is to bridge the gap between different tools in the ecosystem, promoting collaboration, democratization, and optimal runtime efficiency.

Strengths:
- The three use cases (Libraries and Tools Interoperability, Integration of Trainable Modules, Training, and Inference Acceleration) are pertinent to demonstrate the value proposition of Ivy.
- The problem and the solution provided by Ivy and its components are clearly explained.
- Table 1 helps demonstrate the advantage of conversion to specific libraries (i.e., JAX) vs original and other optimized compilations. Although out of the scope of the paper, an in-depth understanding of under which conditions JAX over / underperformers vs. PyTorch Dynamo would be of great interest.

Limitations:
- I suggest adding references to current trends in the interoperability effort since in the current shape Sect. One looks more as if from a white paper rather than a research paper. F.e., ONNX, and Ivy aim to address fragmentation in the DL ecosystem by providing tools for interoperability among different frameworks. While ONNX focuses on a standardized model format for sharing and deployment, Ivy appears to be more focused on enabling code-level interoperability and transpilation between different machine-learning frameworks. However, ONNX can also be used as an intermediate layer to convert code from a source to a target framework, so mentioning it is essential.
- Mentioning the experimental set-up for Table 1 would be beneficial (i.e., how many runs were carried out to retrieve the latency metric, providing both mean and standard deviation, could be added as additional information in Section 3.3 or the Appendix).

Given the strengths of the Ivy framework, enhancing the manuscript with further contextual insights can empower the broader community with a tool to tap into the unique advantages of each deep learning framework. This could help streamline development and deployment, fostering innovation and efficiency and enabling better collaborative code-sharing.

---

### Meta-Review · Area_Chair_YsB9 · 2023-10-28

**Recommendation:** Accept (Poster)
**Confidence:** 4

**Metareview:**

Ivy is the tool that provides a unified frontend that supports multiple backends such as NumPy, JAX, TensorFlow, PyTorch and PaddlePaddle. Thus, it allows the easy integration of code from one framework into another. The reviewer Yxqk confirms the actuality and usefulness of such work for the whole community, but also notices that the other popular tool for transpiling as ONNX is not mentioned. Provided that the reviewer remarks are implemented for the final version, I recommend accepting this paper as this tool can be very helpful for development and deployment of efficient training algorithms.

---

### Decision · Program_Chairs · 2023-10-28

**Decision:**

Accept (Poster)

**Comment:**

We thank the authors for their time and contribution to WANT and we are pleased to share that after the reviewing process the paper has been accepted. Congratulations! We encourage the authors to consider reviewers' feedback for the improvement of the camera-ready version. We hope to see you in person at the workshop and brainstorm on efficient training research together!